# Decrease in α-Globin and Increase in the Autophagy-Activating Kinase ULK1 mRNA in Erythroid Precursors from β-Thalassemia Patients Treated with Sirolimus

**DOI:** 10.3390/ijms242015049

**Published:** 2023-10-10

**Authors:** Matteo Zurlo, Cristina Zuccato, Lucia Carmela Cosenza, Jessica Gasparello, Maria Rita Gamberini, Alice Stievano, Monica Fortini, Marco Prosdocimi, Alessia Finotti, Roberto Gambari

**Affiliations:** 1Department of Life Sciences and Biotechnology, Ferrara University, 44121 Ferrara, Italy; matteo.zurlo@unife.it (M.Z.); cristina.zuccato@unife.it (C.Z.); luciacarmela.cosenza@unife.it (L.C.C.); jessica.gasparello@unife.it (J.G.); 2Center “Chiara Gemmo and Elio Zago” for the Research on Thalassemia, Ferrara University, 44121 Ferrara, Italy; 3Thalassemia Unit, Arcispedale S. Anna, 44121 Ferrara, Italy; gamberinimariarita@gmail.com (M.R.G.); stievano.alice@gmail.com (A.S.); monica.fortini@ospfe.it (M.F.); 4Rare Partners S.r.L. Impresa Sociale, 20123 Milano, Italy; m.prosdocimi@rarepartners.org

**Keywords:** β-thalassemia, autophagy, fetal hemoglobin, α-globin, rapamycin, sirolimus, ULK1

## Abstract

The β-thalassemias are hereditary monogenic diseases characterized by a low or absent production of adult hemoglobin and excess in the content of α-globin. This excess is cytotoxic for the erythroid cells and responsible for the β-thalassemia-associated ineffective erythropoiesis. Therefore, the decrease in excess α-globin is a relevant clinical effect for these patients and can be realized through the induction of fetal hemoglobin, autophagy, or both. The in vivo effects of sirolimus (rapamycin) and analogs on the induction of fetal hemoglobin (HbF) are of key importance for therapeutic protocols in a variety of hemoglobinopathies, including β-thalassemias. In this research communication, we report data showing that a decrease in autophagy-associated p62 protein, increased expression of ULK-1, and reduction in excess α-globin are occurring in erythroid precursors (ErPCs) stimulated in vitro with low dosages of sirolimus. In addition, increased ULK-1 mRNA content and a decrease in α-globin content were found in ErPCs isolated from β-thalassemia patients recruited for the NCT03877809 clinical trial and treated with 0.5–2 mg/day sirolimus. Our data support the concept that autophagy, ULK1 expression, and α-globin chain reduction should be considered important endpoints in sirolimus-based clinical trials for β-thalassemias.

## 1. Introduction

The β-thalassemias are among the most common inherited monogenic hemoglobinopathies worldwide. The genetic alterations leading to the phenotype of β-thalassemias are a variety of autosomal mutations of the gene encoding the β-globin. This causes the absence or low levels of β-globin in an adult hemoglobin (HbA) in erythropoietic cells [1,2,3]. The most severe phenotypes display clinically relevant anemia and require regular transfusion (transfusion-dependent β-thalassemia, TDT, including thalassemia major). Worldwide, more than 300 mutations affecting the expression of the β-globin gene and causing β-thalassemias have now been reported [4]. The low production of β-globin and HbA leads to excess contents of free α-globin, which precipitates and is highly toxic to the erythroid cells [5]. In fact, it is widely accepted that one of the major complications in β-thalassemias is the excess free α-globin chains [6,7,8].

In β-thalassemia patients, the bone marrow is always stimulated by hypoxia and anemia to introduce erythroid precursor cells (ErPCs) into circulation, and these cells undergo early death due to the accumulation of toxic α-globin [4,5]. This phenomenon is a major issue in this pathology because it leads to hemolysis and to a clinical picture called ineffective erythropoiesis, which is harmful even if the patient is regularly transfused. In this respect, rapamycin (sirolimus) [9] is a repositioned drug of great interest.

Two major effects of sirolimus might occur in isolated ErPCs from β-thalassemia patients: (a) the increased production of γ-globin mRNA and HbF [10], and (b) the decrease in free α-globin chains [11]. Concerning the decrease in free α-globin chains, this might be linked to the activation of the γ-globin genes and production of γ-globin to generate the α_2_γ_2_ HbF tetramer [10]. In addition, the decrease in free α-globin chains found in sirolimus-treated erythroid cultures might be due to the activation of the ULK1 (Unc-51–like kinase 1)-dependent increase in autophagy [12], as proposed by Lechauve et al.

In eukaryotes, autophagy is a remarkably conserved degradation mechanism, facilitating the breakdown of superfluous or damaged cellular components (including soluble proteins, aggregated proteins, organelles, macromolecular complexes, and foreign bodies) and their subsequent recycling. The process necessitates the creation of a double-membrane structure known as the autophagosome, which encapsulates the sequestered cytoplasmic material and subsequently merges with lysosomes to break down the cargo and recycle its components [12].

Since in vivo treatments of patients affected with hemoglobinopathies [13,14] and the first clinical trials on β-thalassemia patients [15,16,17,18] have been designed with low sirolimus amounts (in principle not affecting the immune system) [19,20,21,22], we tested whether low sirolimus concentrations (100–200 nM) were able to stimulate ULK-1 dependent autophagy [12,23,24] with an associated decrease in the excess free α-globin chains.

For this purpose, we analyzed autophagy in ErPCs isolated from β-thalassemia patients and treated ex vivo with low doses of sirolimus and ErPCs isolated from β-thalassemia patients recruited in the NCT03877809 clinical trial (SIRTHALACLIN) and treated in vivo with 0.5–2 mg/day sirolimus.

## 2. Results

A possible strategy to verify whether the autophagy process is implicated in α-globin clearance is to compare the autophagy status in the erythroid precursor cells (ErPCs) of healthy donors (cells lacking free α-globin accumulation) versus ErPCs of β-Thalassemia patients.

Therefore, the first point of the study was to verify whether autophagy is operating in erythroid cells from β-Thalassemia patients, where the excess free α-globin chains are a clear pathophysiologic parameter. The second point was to confirm the excess free α-globin chains in β-Thalassemia patients. The third point was to determine the effects of an inducer of fetal hemoglobin (sirolimus) on erythroid cells, both in vitro (using ErPCs isolated from β-Thalassemia patients) and in vivo (studying ErPCs isolated from sirolimus-treated β-Thalassemia patients participating in the NCT03877809 clinical trial) [16].

### 2.1. The Autophagy Program Is Activated in Erythroid Precursor Cells from β-Thalassemia Patients

Figure 1A shows Western blotting analyses suggesting that biochemical markers associated with autophagy are activated in β-thalassemia patients (uncropped version is available in Appendix A). Erythroid precursor cells (ErPCs) from three healthy donors and four β-thalassemia patients (carrying a β^0^/β^0^ genotype) were isolated, and the proteins were analyzed by Western blotting. Figure 1B shows the quantitative analysis of p62 and LC3-I contents, confirming that the major differences were found when the analysis of p62 content was performed. Notably, the p62 content in β-Thalassemia ErPCs was found to be much higher with respect to a healthy donor, and this information is relevant because p62 may be upregulated because of the great quantity of free α-globin protein accumulated in these cells, which must be transported into autophagosomes (presumably by p62 itself).

As ULK-1 is deeply involved in autophagy (especially in the case of excess α-globin chain production by erythroid cells, as demonstrated by Lechauve et al. [12]), the expression of ULK-1 mRNA and α-globin mRNA was analyzed using RT-qPCR. Figure 1C,D show the changes in ULK-1 mRNA (Figure 1C) and α-globin mRNA (Figure 1D) in the β-thalassemia ErPCs, compared with ErPCs from healthy donors. An HPLC analysis of hemoglobin was also performed on cell lysates to verify the excess free α-globin in β-thalassemia-cultured cells, confirming the expected accumulation of free α-globin chains in ErPCs of β-Thalassemia patients, while the healthy donor does not present any peak in the HPLC chromatogram corresponding to free α-globin chains (Figure 1E).

### 2.2. Sirolimus Potentiates Autophagy in Erythroid Precursor Cells from β-Thalassemia Patients

We tested several concentrations of sirolimus (ranging from 200 nM to 20 μM, respectively) in order to know the working concentration necessary to obtain a biological effect. We also further investigated ULK1 gene expression (ULK1 complex drives formation of the phagophore and thereby autophagy initiation) through RT-qPCR and p62 levels using the Western blot technique. The Western blot analysis showed that p62 content decreased in sirolimus-treated ErPCs (Figure 2A), especially after five days of treatment (uncropped version of the Western blot is showed in Appendix A). The ULK1 mRNA content was found to increase following sirolimus treatment (Figure 2B). Also, 200 nM sirolimus and two days of treatment were sufficient to induce ULK-1 gene expression.

The data on p62 were of interest, as the p62 protein is responsible for misfolded protein capturing and addressing to autophagosomes for subsequent degradation. The decrease in the cellular content of p62 under treatment with autophagy inducers is considered to be evidence of the activation of the autophagy process. In this regard, it is particularly intriguing to find a correlation with data obtained through the HPLC analysis (Figure 2C), showing a correlation between the decrease in p62 levels and free α-globin chain after 2 and 5 days of treatment with sirolimus. In fact, after 5 days of treatment, each tested concentration of sirolimus produced a marked decrease in p62 and free α-globin chains levels (Figure 2A–C).

Increased expression of γ-globin was verified following sirolimus treatment and, as expected, the HbF percentage was significantly higher in treated ErPCs (despite the great variability among patients) (Figure 2D, upper side of the panel).

In conclusion, the reduction in free α-chain values due to the sirolimus treatment of β-Thalassemia ErPCs in vitro (Figure 2C,D, middle side of the panel) might be caused by two concurrent molecular events, i.e., the induction of HbF (Figure 2D, upper side of the panel) and the increase in autophagy-associated ULK-1 gene expression (Figure 2D, lower side of the panel).

### 2.3. Sirolimus-Induced Autophagy Reduces Accumulation of Insoluble α-Chains in Membrane Skeletons of Cultured ErPCs

A key experiment for understanding whether the sirolimus-mediated induction of autophagy (see Figure 1 and Figure 2) affects α-globin clearance was performed by analyzing the content of insoluble α-globin in lysed membranes of β-Thalassemia ErPCs. In fact, it is well known that excess α-globin produces insoluble α-globin aggregates that are toxic for cells and cause the premature cell death of ErPCs (the major cause of ineffective erythropoiesis). For this reason, the α-globin aggregates are stocked in “inclusion bodies” that are usually linked to the cytoplasmatic membrane. Accordingly, if sirolimus is able to reduce the content of free α-globin through γ-globin pairing and/or autophagy induction, the content of insoluble α-globin in inclusion bodies should be lower. To verify this, we lysed β-Thalassemia ErPCs in order to remove all soluble α-globin contents (analyzed by HPLC), and we washed the membrane skeletons several times before proceeding with insoluble protein extraction as explained in the methods section. By following this protocol, we could check the content of accumulated α-globin protein in membrane skeletons of cultured ErPCs (Figure 3).

Representative data shown in Figure 3 indicate that even low concentrations of sirolimus can induce α-globin clearance and lower the accumulation of insoluble α-globin aggregates (uncropped Western blot images are shown in Appendix A). These data suggest that autophagy induction may play a key role in the biological effect of sirolimus on α-globin clearance in β-Thalassemias, in addition to γ-globin mRNA induction and HbF production.

We analyzed five different patients treated with sirolimus 200 nM in vitro and found a significant increase in ULK1 gene expression, with a concomitant decrease in p62 adaptor protein together with a decrease in soluble and insoluble α-chain accumulation in erythroid cells (Figure 4).

### 2.4. Sirolimus Treatment In Vivo Leads to a Lower Accumulation of Free α-Chains

To sustain the hypothesis derived from experiments conducted in vitro, we retrospectively tested the bio-banked RNA samples derived from the SIRTHALACLIN clinical trial (NCT03877809) for ULK1 mRNA content (Figure 5).

The RT-qPCR analysis confirmed a clear increase in ULK-1 gene expression in ErPCs isolated from β-Thalassemia patients enrolled in the SIRTHALACLIN clinical study after 3 months of daily sirolimus administration. Once again, the ULK-1 mRNA content seems to be higher when ErPCs are exposed to sirolimus (this time in vivo) and, if this is correlated with autophagy induction as shown above, could be an important part of sirolimus biological effects.

## 3. Discussion

The employment of mTOR inhibitors for the treatment of β-thalassemia and sickle-cell diseases is supported by several laboratory investigations, pre-clinical studies, and clinical data [13,25,26,27]. In agreement with this conclusion, the in vivo effects of mTOR inhibition were reported in a pre-clinical study by Wang et al. [28], who found that sirolimus improved anemia and reduced organ damages in a murine model of sickle cell disease. Accordingly, Khaibullina et al. reported that sirolimus increased HbF in another model of sickle cell mice [29].

Concerning clinical data, Al-Khatti and Alkhunaizi reported a study on two patients with SCD who were treated with sirolimus as part of their post-kidney transplantation immunosuppressive regimen. During this study, they observed an increase in HbF levels in the treated patients. In addition, there was a further increase in HbF levels when sirolimus and hydroxycarbamide were combined [14]. The conclusion of this study is in line with the paper by Gaudre et al. [13] reporting improved HbF production with everolimus, another mTOR inhibitor. This study was also conducted on a kidney transplant recipient patient with SCD. These studies support the recent Orphan Drug Designation (ODD) of sirolimus by the European Medicinal Agency (EMA) for the treatment of β-thalassemias (code EU/3/15/1585) and SCD (code EU/3/17/1970). The ODD of sirolimus as an HbF inducer in β-thalassemias and SCD was also obtained by the U.S. Food and Drug Administration (FDA). Clinical data of the possible use of sirolimus in β-thalassemias have been obtained by two clinical trials, NCT03877809 and NCT04247750, focusing on the in vivo effects of sirolimus in treated β-thalassemia patients [16,17]. A first report concerning the NCT03877809 trial (Sirthalaclin) was recently published by Zuccato et al. [18], who demonstrated increased content of γ-globin mRNA and increased production of HbF in ErPCs isolated from sirolimus-treated β-thalassemia patients.

In addition to the effects of rapamycin on HbF, this drug might be of great interest in protocols for β-thalassemias because it reduces the α-globin/β-globin imbalance by activating autophagy. This was proposed by Lechauve et al. (2019) who reported the ULK1 (Unc-51–like kinase 1)-dependent increase in autophagy in a mouse thalassemia model system, leading to a sharp decrease in free α-globin chains [12].

Our research group has recently obtained K562 cellular clones that are able to hyper-express α-globin protein [30]. In this work we have demonstrated that the accumulation of toxic α-globin is able to induce apoptosis proportionally to its production/accumulation rate. Interestingly, we found that autophagy is spontaneously triggered by the accumulation of α-globin protein, and cellular clones that produce higher levels of the protein seem to have higher levels of p62 adaptor protein, probably in order to manage α-globin clearance. In the present work, we obtained similar evidence by analyzing the p62 content in healthy donor cells compared to β-thalassemia erythroid cells, confirming that accumulation of α-globin protein requires higher levels of p62.

Moreover, we demonstrated that sirolimus treatment of cultured β-thalassemia ErPCs in vitro can trigger/potentiate the autophagy process by reducing p62 levels and concurrently reduce free α-globin chains present in the cytoplasm and insoluble α-globin aggregates content in membrane skeletons. These data obtained on human β-thalassemia ErPCs confirm and strengthen the data obtained by Lechauve et al. [12] in a β-thalassemia mouse model. In addition, we obtained similar results on autophagy induction using a cellular treatment protocol based on much lower concentrations of sirolimus, approaching the therapeutic blood concentrations reached in clinical trials, such as NCT03877809 and NCT04247750.

This study is a contribution toward further understanding the in vivo sirolimus-based mechanism of action. A limit of this study is the fact that the link between autophagy and the proteasome machinery was not investigated. As p62 is involved in both autophagy and proteasome activity [31,32], this issue should be carefully analyzed in the future. In this respect, proteasome has been suggested to be implicated in decongesting erythroid precursors and mature red blood cells from excess hemoglobin [33,34,35]. This observation reinforces the interest in analyzing the effects of sirolimus on proteasome [36,37]. This study should be considered in the future since several reports have suggested that rapamycin allosterically impacts proteasome function [37].

The most important result of our study is the demonstration that low concentrations of sirolimus (100 and 200 nM) were able to induce ULK1 and to reduce p62, strongly suggesting the activation of autophagy in erythroid precursor cells (ErPCs) from β-thalassemia patients (Figure 4). This was confirmed in our study using ErPCs from sirolimus-treated β-thalassemia patients participating in the NCT03877809 (Sirthalaclin) clinical trial. In these ErPCs, ULK1 was found to be upregulated (Figure 5A), and the excess free α-globin chains were strongly reduced (Figure 5B).

In our opinion, our results on ULK-1 expression in ErPCs isolated from sirolimus-treated patients are of interest since the role of ULK-1 in the pathophysiology of thalassemias is a topic of active interest [12] and translational applications [38].

In conclusion, the presented data support the hypothesis that, in addition to the induction of fetal hemoglobin, sirolimus reduces the excess free α-globin chains by activating autophagy via ULK-1 gene expression, both in vitro and in vivo. A reduction in the excess insoluble free α-globin chains and the expression of autophagy-related genes (including ULK-1) should be considered, in addition to γ-globin and HbF production, as endpoints in sirolimus-based clinical trials for β-thalassemias, such as the NCT03877809 (2019) and NCT04247750 (2020) clinical trials.

## 4. Materials and Methods

### 4.1. Culture and Treatment of Human Erythroid Precursor Cells

The two-phase liquid culture procedure was employed as previously described [18,39,40,41,42]. Mononuclear cells were isolated from peripheral blood samples of β-thalassemia patients through centrifugation using Lympholyte® (Cederlane, Burlington, ON, Canada). After isolation, the mononuclear cells were washed three times in PBS solution and seeded in an α-minimal essential medium (α-MEM, Sigma-Aldrich, St. Louis, MO, USA) supplemented with 10% FBS (Celbio, Milano, Italy), 1 µg/mL cyclosporine A (Sandoz, Basel, Switzerland), and a 10% conditioned medium from a CM5637 bladder carcinoma cell line culture. The cultures were incubated in a standard atmosphere of 5% CO_2_ at 37 °C. After 7 days in this phase I culture, non-adherent cells were harvested from the flask, washed with PBS, and then cultured in the phase II medium, composed of α-MEM medium, 30% FBS, 1% deionized bovine serum albumin (BSA) (Sigma-Aldrich, St. Louis, MO, USA), 10^−5^ M β-mercaptoethanol (Sigma-Aldrich, St. Louis, MO, USA), 2 mM L-glutamine (Sigma-Aldrich, St. Louis, MO, USA), 10^−6^ M dexamethasone (Sigma-Aldrich, St. Louis, MO, USA), 1 U/mL human recombinant erythropoietin (EPO) (Tebu-bio, Magenta, Milano, Italy), and stem cell factor (SCF, BioSource International, Camarillo, CA, USA) at the final concentration of 10 ng/mL. After five days of expansion in the phase II medium, cells were plated at 2 mln cells/mL concentration and treated with sirolimus (Sigma-Aldrich, St. Louis, MO, USA) resuspended in EtOH for additional 5 days.

### 4.2. RNA Extraction and RT-qPCR Analysis

The total cellular RNA was extracted from ErPCs by using TRI Reagent^®^ (Sigma-Aldrich, St. Louis, MO, USA) and following the manufacturer’s instructions. The isolated RNA was washed once with cold 75% ethanol, dried, and dissolved in a 10 μL nuclease-free water before use.

For gene expression analysis, 500 ng of total RNA was reverse-transcribed by using the TaqMan^®^ Reverse Transcription Reagents and random hexamers (Applied Biosystems, Life Technologies, Carlsbad, CA, USA). Quantitative real-time PCR assay, to quantify the expression of the globin genes, was carried out using two different reaction mixtures, the first one containing α, β, γ-globin probes and primers, and the second one containing GAPDH, RPL13A, β-actin probes and primers. In parallel, ULK1 gene expression was quantified and normalized with the same reference sequences employed for globin gene expression normalization. The primers and probes used are listed in Table 1.

Each reaction mixture contained 1× TaKaRa Ex Taq^®^ DNA Polymerase (Takara Bio Inc., Shiga, Japan), 300 nM forward and reverse primers, and the 200 nM probes (Integrated DNA Technologies, Castenaso, Italy). The assays were carried out using CFX96 Touch Real-Time PCR System (Bio-Rad, Hercules, CA, USA). After an initial denaturation at 95 °C for 1 min, the reactions were performed for 50 cycles (95 °C for 15 s, 60 °C for 60 s). Data were analyzed by employing the CFX manager software (Bio-Rad, Hercules, CA, USA). To compare the gene expression of each template amplified, the ΔΔCt method was used [39,42].

### 4.3. HPLC Analyses of Hemoglobin

To evaluate free α-globin variations, HPLC analysis was carried out using fresh lysates of ErPCs. The ErPCs were centrifuged at 1500 rpm for 5 min and washed with PBS. The pellets were then lysed in a predefined volume of water for HPLC (Sigma-Aldrich, St. Louis, MO, USA) in order to obtain protein extracts suitable for HPLC analysis. Lysed pellets in water for HPLC were incubated 20 min in ice and then pelleted at 16,000 rpm for 15 min at 4 °C. Finally, protein extracts were recovered in a pre-chilled 1.5 mL tube and injected in the HPLC system. Hemoglobin analysis was performed by loading the protein extracts into a PolyCAT-A cation exchange column and then eluted in a sodium-chloride-BisTris-KCN aqueous mobile phase using HPLC Beckman Coulter instrument System Gold 126 Solvent Module-166 Detector [18]. The reading was performed at a wavelength of 415 nm, and a commercial solution of purified human HbF (Sigma-Aldrich, St. Louis, MO, USA) extracts was used as standard.

### 4.4. Western Blotting of Soluble Fractions

To analyze soluble fractions of ErPCs, we employed the same protein extracts prepared for HPLC analysis as explained above. Protein concentration was determined using PierceTM BCA Protein Assay Kit (Thermo Fisher, Waltham, MA, USA) to run the gels. Then, 20 μg of cytoplasmic extracts was denatured for 5 min at 98 °C in SDS sample buffer (Cell Signaling Technology, Danvers, MA, USA) and loaded on hand-casted SDS-PAGE 15% gel (10 cm × 8 cm) in Tris-glycine Buffer (Bio-Rad, Hercules, CA, USA). Spectra prestained multicolor protein ladder (Thermo Fisher, Waltham, MA, USA) was used as the standard to determine molecular weight (size range 10–260 kDa). An electrotransfer to a 0.2 μm pore size nitrocellulose membrane (Thermo Fisher, Waltham, MA, USA) was performed overnight at 360 mA and 4 °C in standard Tris-Glycine-MeOH transfer buffer. The membranes were stained with Ponceau S Solution (Sigma-Aldrich, St. Louis, MO, USA) to verify the efficiency of the transfer, washed with TBS-T (Cell Signaling Technology, Danvers, MA, USA), and then blocked with standard blocking buffer (5% milk in TBST) for 1 h at room temperature. The membranes were washed three times with TBS-T and incubated with the primary antibody in 10 mL of 5% BSA in TBS-T with gentle shaking overnight at 4 °C. The next day, the membranes were washed three times with TBST and incubated in 10 mL of blocking buffer, with gentle shaking for 2 h at room temperature, with an appropriate HRP-conjugated secondary antibody. Finally, after three washes with TBS-T, each membrane was incubated with 5 mL LumiGLO® detection working solution (Cell Signaling Technology, Danvers, MA, USA) and exposed to X-ray film (Pierce, Euroclone S.p.A., Pero, MI, Italy). In order to re-probe the membranes, they were stripped using the RestoreTM Western Blot Stripping Buffer (Thermo Fisher, Waltham, MA, USA), incubating the membranes for 30 min at 37–42 °C with moderate agitation. Blots images were acquired and analyzed using Bio-Rad Image Lab Software (Bio-Rad, Hercules, CA, USA). A list of employed primary and secondary antibodies is presented in Table 2.

### 4.5. Western Blotting of Insoluble Fractions

The protocol followed for the detection of insoluble α-chains started with standard cytoplasmatic extracts obtained from cultured ErPCs (as described for HPLC samples). Once the cytoplasmatic extracts were separated from lysed membranes, several washing steps in hypotonic water were used to clean membrane skeletons pellets from residual hemoglobin (at least five washing steps in PBS). At the end of the washing steps, membranes were extracted from the pellet with addition of Sodium Borate extraction buffer (Sodium Borate 68 mM-pH 8.2 and 0.1% Tween-20) for 20 min, keeping the samples in ice and vortexing every 5 min. Finally, the samples were centrifuged at 4 °C for 30 min at 16,000× *g*; extracted insoluble proteins pellets were resuspended in an adequate volume of a 1× SDS sample buffer, boiled for 5 min, and well-vortexed and stocked at −80 °C [33]. As reported in Table 3, in this case, data were normalized with GAPDH, as it was more present inside membrane protein extracts with respect to standard cytoplasmatic extracts. After sample preparation, the Western blot procedure was the same as described above in Western blotting of soluble fractions.

### 4.6. Statistical Analysis

All the data were normally distributed and presented as mean ± S.D. Statistical differences between groups were compared using Prism Software v9.02 and employing two-tail paired *t*-test. Statistical differences were considered significant when *p* < 0.05 (*) and highly significant when *p* < 0.01 (**).

## Figures and Tables

**Figure 1 ijms-24-15049-f001:**
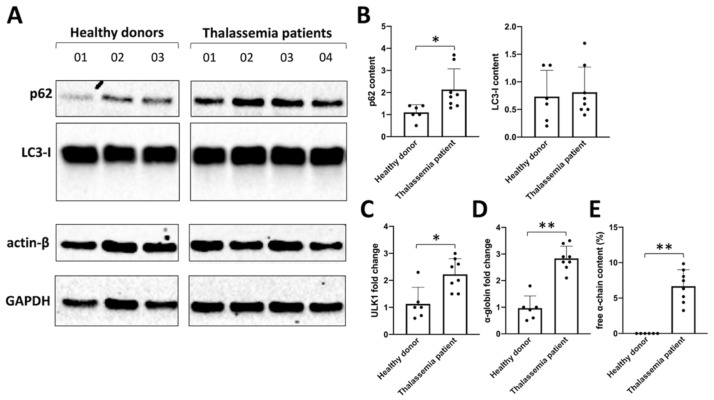
(**A**) Western blot comparing p62 and LC3 levels in ErPCs isolated from three healthy donors versus four Thalassemia patients and relative densitometry data set (**B**). ULK1 mRNA content (**C**) and α-globin mRNA content (**D**) were analyzed by RT-qPCR, while free α-chain content (**E**) was measured by HPLC analysis. (*): *p* < 0.05 (significant); (**): *p* < 0.01 (highly significant).

**Figure 2 ijms-24-15049-f002:**
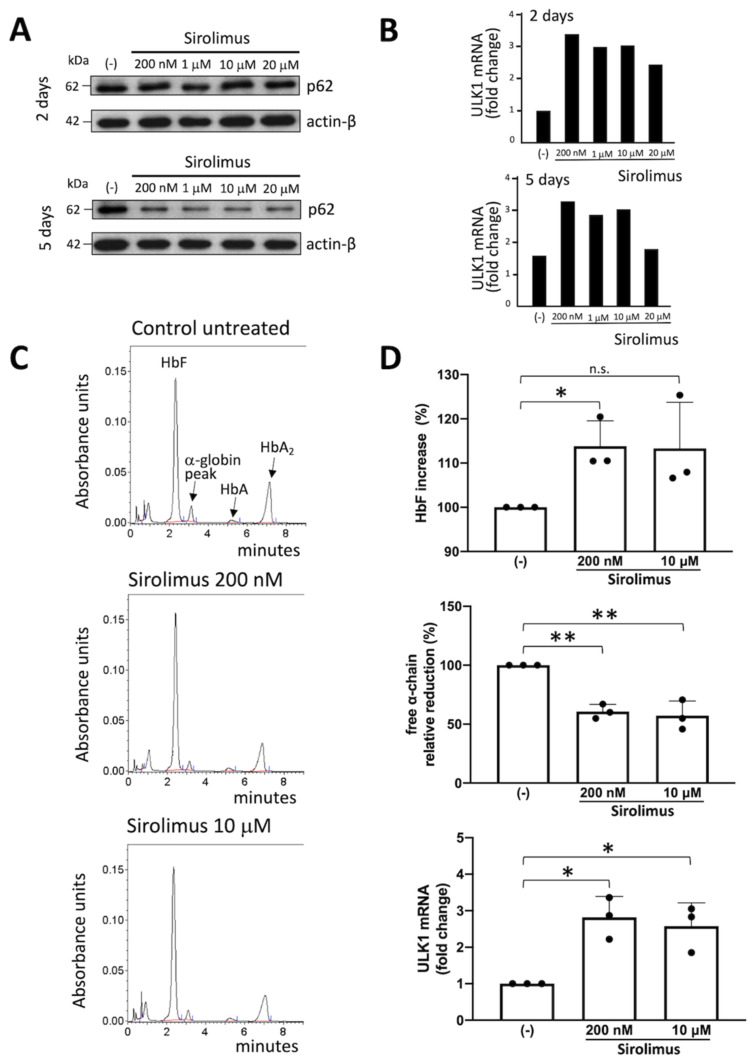
(**A**–**C**) Representative experiments of the effects of treatment of ErPCs from a homozygous β^0^ thalassemia patient with the indicated concentrations of sirolimus. (**A**) Western blotting showing the effects on p62 and β-actin (2 and 5 days of treatment). (**B**) ULK-1 mRNA content analyzed using RT-qPCR after 2 and 5 days of treatment. (**C**) HPLC analysis after 5 days of treatment. (**D**) Effects of treatment with 200 nM and 10 µM sirolimus on ErPCs from three β-thalassemia patients on HbF increase, decrease in the free α-globin peak and increase in ULK-1 gene expression. (n.s.): not significant; (*): *p* < 0.05 (significant); (**): *p* < 0.01 (highly significant).

**Figure 3 ijms-24-15049-f003:**
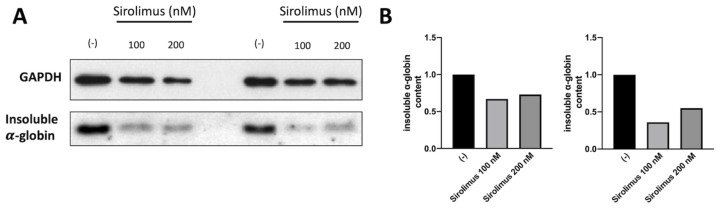
Analysis of insoluble α-globin content in lysed membranes of ErPCs isolated from two different Thalassemia patients with β0/β+ genotype and treated for six days with Sirolimus (**A**). Related densitometry data (**B**). In both patients, Sirolimus treatment reduced insoluble α-globin accumulation in analyzed membrane skeletons.

**Figure 4 ijms-24-15049-f004:**
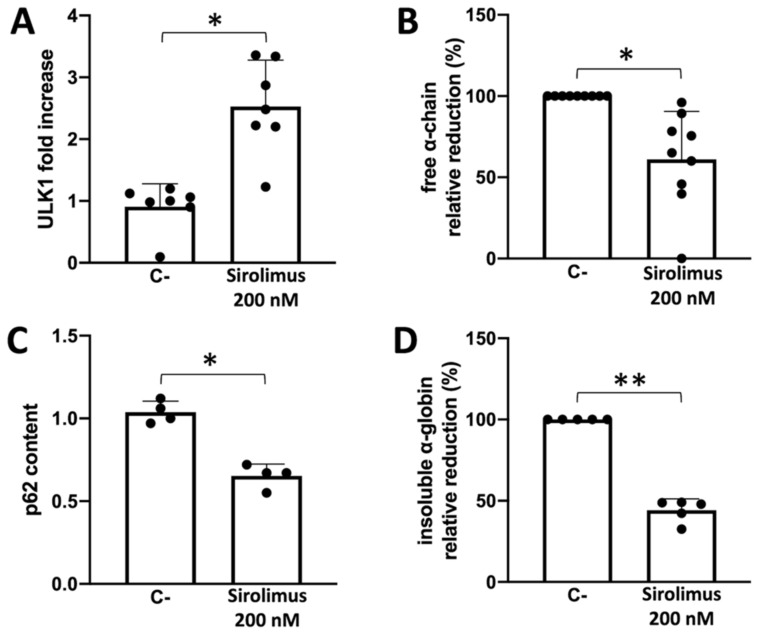
Summary showing autophagy correlation with α-globin clearance in ErPCs isolated from β-Thalassemia patients and cultured in the presence of Sirolimus 200 nM or alone (mean ± SD; N = 5). (**A**) Mean ULK1 mRNA increase analyzed by RT-qPCR and normalized on the patient with lower ULK1 levels. (**B**) Free α-globin chains reduction relative to each patient baseline level was determined by HPLC analyses. (**C**) p62 reduction in ErPCs cellular lysates was analyzed by Western blot. (**D**) Relative reduction of insoluble α-globin content in ErPCs membrane skeletons evaluated using the Western blot technique. (*): *p* < 0.05 (significant); (**): *p* < 0.01 (highly significant).

**Figure 5 ijms-24-15049-f005:**
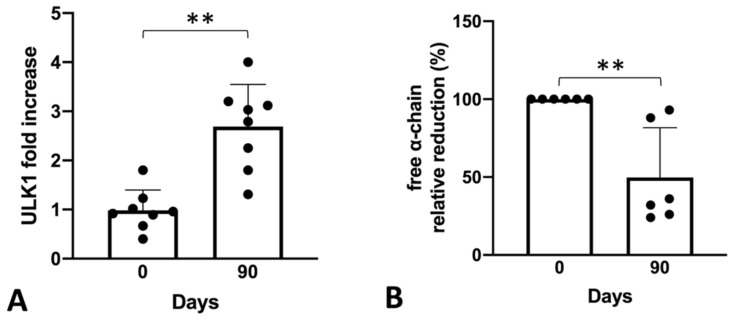
Summary showing correlation of autophagy and free α-globin clearance in bio-banked ErPCs isolated from β-Thalassemia patients enrolled in SIRTHALACLIN clinical trial before and after 3 months of Sirolimus treatment in vivo (mean ± SD; N = 7). The mean ULK1 mRNA increase was analyzed by RT-qPCR and normalized on the patient with a lower detected level, (**A**) and free α-globin chains reduction relative to each patient baseline level (**B**) was determined using HPLC analyses. (**): *p* < 0.01 (highly significant).

**Table 1 ijms-24-15049-t001:** List of primers and probes with related sequences used to perform RT-qPCR analyses.

Primers and Probes	Sequences
primer forward α-globin	5′-CGACAAGACCAACGTCAAGG-3′
primer reverse α-globin	5′-GGTCTTGGTGGTGGGGAAG-3′
probe α-globin	5′-HEX-ACATCCTCTCCAGGGCCTCCG-BFQ-3′
primer forward ULK1	5′-CTACCTGGTTATGGAGTACTGC-3′
primer reverse ULK1	5′-GGAAGAGCCTGATGGTGTC-3′
probe ULK1	5′-FAM-CGACTACCT/ZEN/GCACGCCATGC-BFQ-3′
primer forward RPL13A	5′-GGCAATTTCTACAGAAACAAGTTG-3′
primer reverse RPL13A	5′-GTTTTGTGGGGCAGCATACC-3′
probe RPL13A	5′-HEX-CGCACGGTCCGCCAGAAGAT-BFQ-3′
primer forward ACTB	5′-ACAGAGCCTCGCCTTTG-3′
primer reverse ACTB	5′-ACGATGGAGGGGAAGACG-3′
probe ACTB	5′-Cy5-CCTTGCACATGCCGGAGCC-BRQ-3′
primer forward GAPDH	5′-ACATCGCTCAGACACCATG-3′
primer reverse GAPDH	5′-TGTAGTTGAGGTCAATGAAGGG-3′
probe GAPDH	5′-FAM-AAGGTCGGAGTCAACGGATTTGGTC-BFQ-3′

**Table 2 ijms-24-15049-t002:** Western blot primary and secondary antibodies employed for detection of protein present in soluble fraction of ErPCs.

Target	Primary Antibody	Cat. n.	Secondary Antibody	Cat. n.
p62	Rabbit anti-p62/SQSTM1 (Sigma-Aldrich, St. Louis, MO, USA)	P0067	Mouse Anti-rabbit IgG HRP (Cell Signaling Technology, Danvers, MA, USA	7074
LC3	Rabbit anti-LC3B (Sigma-Aldrich, St. Louis, MO, USA)	L7543	Mouse Anti-rabbit IgG HRP (Cell Signaling Technology, Danvers, MA, USA)	7074
β-actin	Rabbit anti-β-actin (Cell Signaling Technology, Danvers, MA, USA)	4967	Mouse Anti-rabbit IgG HRP (Cell Signaling Technology, Danvers, MA, USA	7074
GAPDH	Mouse anti-GAPDH (Thermo Fisher, Waltham, MA, USA)	MA1-16783	Goat Anti-mouse IgG HRP (Thermo Fisher, Waltham, MA, USA)	32430

**Table 3 ijms-24-15049-t003:** Western blot primary and secondary antibodies employed for detection of α-globin in insoluble fraction of ErPCs.

Target	Primary Antibody	Cat. n.	Secondary Antibody	Cat. n.
α-globin	Mouse anti-hemoglobin α (D-4) (Santa Cruz Biotechnology, Dallas, TX, USA)	sc-514378	Goat Anti-mouse IgG HRP (Thermo Fisher, Waltham, MA, USA)	32430
GAPDH	Mouse anti-GAPDH (Thermo Fisher, Waltham, MA, USA)	MA1-16783	Goat Anti-mouse IgG HRP (Thermo Fisher, Waltham, MA, USA)	32430

## Data Availability

Materials and further information on the data will be freely available upon request to the corresponding authors.

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
