# Peer review of "Decrease in α-Globin and Increase in the Autophagy-Activating Kinase ULK1 mRNA in Erythroid Precursors from β-Thalassemia Patients Treated with Sirolimus"

_ijms, 2023, doi:10.3390/ijms242015049_

Round 1
Reviewer 1 Report
In their manuscript Zurlo et al. present information regarding the effect of sirolimus upon autophagy in beta-thalassemia. The manuscript is important to understand how sirolimus works in the treatment of beta-thalassemia. Nonetheless, there is a point that needs to be addressed before the publication of this work.
Since p62 is a link between autophagy and the proteasome machinery (https://doi.org/10.1186/s11658-016-0031-z; 10.5483/BMBRep.2020.53.1.283), it would make the study more complete if proteasomal activity and the levels of proteasome subunits were also studied. Moreover, proteasome has been suggested to be implicated in decongesting erythroid precursors and mature red blood cells from the excess of hemoglobin (10.1182/blood-2011-12-397729; 10.3389/fphys.2022.907444; 10.3390/membranes11090716). Could the authors check some aspects of proteostasis? If not, I believe that they should at least put some pieces of information (bibliographical and hypothesis-based) in their manuscript before its acceptance. Of course, the fact that sirolimus might block proteasome should be also discussed (10.1124/mol.112.083873).
Only minor mistakes were spotted regarding the use of English.
Author Response
Reply to Reviewer 1
In their manuscript Zurlo et al. present information regarding the effect of sirolimus upon autophagy in beta-thalassemia. The manuscript is important to understand how sirolimus works in the treatment of beta-thalassemia. Nonetheless, there is a point that needs to be addressed before the publication of this work.
Since p62 is a link between autophagy and the proteasome machinery (https://doi.org/10.1186/s11658-016-0031-z; 10.5483/BMBRep.2020.53.1.283), it would make the study more complete if proteasomal activity and the levels of proteasome subunits were also studied. Moreover, proteasome has been suggested to be implicated in decongesting erythroid precursors and mature red blood cells from the excess of hemoglobin (10.1182/blood-2011-12-397729; 10.3389/fphys.2022.907444; 10.3390/membranes11090716). Could the authors check some aspects of proteostasis? If not, I believe that they should at least put some pieces of information (bibliographical and hypothesis-based) in their manuscript before its acceptance. Of course, the fact that sirolimus might block proteasome should be also discussed (10.1124/mol.112.083873).
Answer. We are grateful for the reviewer’s suggestion and work very useful for improving the manuscript. We agree with all suggestions and we thank for the references suggested. In order to consider all the points raised the following sentence has been added: “This study is a contribution to further understand the in vivo sirolimus-based mechanism of action. A limit of this study is the fact that the link between autophagy and the proteasome machinery has not been investigated. As p62 is involved …… should be considered in the future, on the basis of several reports suggesting that rapamycin allosterically inhibits the proteasome machinery [37]” (pages 7 and 8, lines 241-249). In order to sustain the sentence, the references 31-37 has been added.

Reviewer 2 Report
It is a well designed research study. The group of authors have an experience with other published papers regarding sirolimus and thalassemias
Introduction summarizes the current knowledge, methods and materials describe in detail the project, results are given adequately. In discussion, it is presented the research work for sirolimus effect in thalassemic patients. Most of this work is part of research field of the authors that has already been published
Minor editing
Author Response
Reply to Reviewer 2
It is a well designed research study. The group of authors have an experience with other published papers regarding sirolimus and thalassemias.
Introduction summarizes the current knowledge, methods and materials describe in detail the project, results are given adequately. In discussion, it is presented the research work for sirolimus effect in thalassemic patients. Most of this work is part of research field of the authors that has already been published.
Answer. We thank the reviewer for her/his positive comments.

Reviewer 3 Report
The authors have previously published a clinical proof-of-principle study (reference 18) demonstrating the effect of sirolimus on gamma globin production. The clinical phenotype of thalassemia largely reflects the imbalance between alpha and beta chain production, and increased gamma globin can potentially mitigate the clinical phenotype by binding excess alpha. In addition, activation of autophagy provides an alternative mechanism for clearance of excess alpha globin chains.
In the present study, the authors employ an in vitro erythroid precursor model to characterize the mechanistic basis for the results in the studies cited above. The studies are carefully performed, the results are appropriately described, and the logic underlying study design is very clear.
The role of ULK1 in the pathophysiology of thalassemia is a topic of active interest currently, so this paper is very much up to date and in the mainstream.
Author Response
Reply to Reviewer 3.
The authors have previously published a clinical proof-of-principle study (reference 18) demonstrating the effect of sirolimus on gamma globin production. The clinical phenotype of thalassemia largely reflects the imbalance between alpha and beta chain production, and increased gamma globin can potentially mitigate the clinical phenotype by binding excess alpha. In addition, activation of autophagy provides an alternative mechanism for clearance of excess alpha globin chains.
In the present study, the authors employ an in vitro erythroid precursor model to characterize the mechanistic basis for the results in the studies cited above. The studies are carefully performed, the results are appropriately described, and the logic underlying study design is very clear.
The role of ULK1 in the pathophysiology of thalassemia is a topic of active interest currently, so this paper is very much up to date and in the mainstream.
Answer. First of all, we thank the reviewer for the positive comments. We fully agree that “The role of ULK1 in the pathophysiology of thalassemia is a topic of active interest currently”. We further reinforced this concept by adding the short sentence “In our opinion, our results on ULK-1 expression in ErPCs isolated from sirolimus treated patients are of interest, since the role of ULK-1 in the pathophysiology of thalassemia is a topic of active interest [12] and translational application [38]”, citing the very interesting paper, recently published by Keith et al. (2023).(page8, lines 257-259.

Round 2
Reviewer 1 Report
The authors addressed adequately all my comments so I believe this paper can be accepted for publication in its current form.
Only minor mistakes were noticed regarding the use of English.
Author Response
Reply to reviewer 2
We thank the reviewer for her/his positive comments.
We have checked the manuscript fixing English mistakes.